# Sintilimab with two cycles of chemotherapy for the treatment of advanced squamous non-small cell lung cancer: a phase 2 clinical trial

Mina Zhang[1], Guowei Zhang[1], Yuanyuan Niu[1], Guifang Zhang[2], Yinghua Ji[3], Xiangtao Yan[1], Xiaojuan Zhang[1], Qichuan Wang[4], Xiaohui Jing[5], Junsheng Wang[6], Zhiyong Ma[1] & Huijuan Wang ®[1] ✉

This was a single-arm, multicenter phase 2 clinical trial (ChiCTR1900021726) involving advanced squamous non-small cell lung cancer (sq-NSCLC) patients undergoing 2 cycles of nab-paclitaxel/carboplatin and sintilimab (anti-PD-1), followed by sintilimab maintenance therapy. The median progression-free survival (PFS) was 11.4 months (95% CI: 6.7-18.1), which met the pre-specified primary endpoint. Secondary endpoints included objective response rate reaching 70.5% and a disease control rate of 93.2%, with a median duration of response of 13.6 months [95% CI: 7.0–not evaluable (NE)]. The median overall survival was 27.2 months (95% CI: 20.2–NE) with treatment-related adverse events grades ≥3 occurring in 10.9% of patients. Predefined exploratory endpoints comprised relationships between biomarkers and treatment efficacy, and the association between circulating tumor DNA (ctDNA) dynamics and PFS. Biomarker analysis revealed that the breast cancer gene 2, BMP/Retinoic Acid Inducible Neural Specific 3, F-box/WD repeat-containing protein 7, tyrosine-protein kinase KIT and retinoblastoma 1 abnormalities led to shorter PFS, while ctDNA negative at baseline or clearance at 2 cycles of treatment was associated with longer PFS (18.1 vs. 4.3 months). Taken together, sintilimab in combination with 2 cycles of nab-paclitaxel/carboplatin treatment produced encouraging PFS and better tolerability as first-line treatment for advanced sq-NSCLC.

Lung squamous cell carcinoma accounts for about 25%–30% of non-small cell lung cancer (NSCLC)[1,2]. Previous studies found that squamous NSCLC (sq-NSCLC) is the major histopathological cancer type in males, with a 5.7 times higher incidence in males than in females in Chinese non-smokers[3]. Due to its specific clinicopathological characteristics and lack of targeted therapeutic drugs, the development of treatment for this condition has stalled, and survival after conventional chemotherapy is disappointingly limited[4]. Immune checkpoint

[1]Department of Medical Oncology, The Affiliated Cancer Hospital of Zhengzhou University/Henan Cancer Hospital, 127 Dongming Rd, Zhengzhou 450003, China. [2]Department of Medical Oncology, Xinxiang Central Hospital, 56 Jinsui Rd, Xinxiang 453000, China. [3]Department of Medical Oncology, The First Affiliated Hospital of Xinxiang Medical University, 88 Jiankang Rd, Xinxiang 453199, China. [4]Department of Medical Oncology, The Second People's Hospital of Nanyang, 66 Jianshe Rd, Nanyang 473000, China. [5]Department of Medical Oncology, The First People's Hospital of Pingdingshan, 117 Youyue Rd, Pingdingshan 467099, China. [6]Department of Medical Oncology, Anyang Cancer Hospital, 2 N Huanbin Rd, Anyang 455001, China. ✉e-mail: 1638561588@163.com

inhibitors (ICIs) have produced promising therapeutic effects in recent years and have become the standard first-line therapy for advanced lung squamous cell carcinoma. However, the optimal model for immunotherapy still needs to be further explored. Monotherapy[5–7] or dual immunotherapy (chemo-free) trials[8] have indicated that the early survival curves overlapped, suggesting that some patients might not have benefited from immunotherapy in the first few weeks. Multiple large-scale phase 3 clinical trials[9–11], combining standard 4–6 cycles of chemotherapy with immunotherapy, have revealed that patients in the immunotherapy group experienced significantly longer survival times compared to the controls. However, a significant proportion of patients also exhibited poor tolerability to the drugs.

The CheckMate 9LA study conducted 2-cycle of chemotherapy in combination with dual immunotherapy compared to chemotherapy alone in advanced NSCLC patients[12]. It was found that the median overall survival (OS) of the immunotherapy group improved significantly compared to the chemotherapy group [hazard ratio (HR) = 0.66], and the difference of survival between the experimental and chemotherapy group was more significant than dual immunotherapy in CheckMate227[13]. The survival curves for progression-free survival (PFS) and OS were apparently separated at an early stage, indicating that chemotherapy could enhance the early response to immunotherapy. Several previous studies have indeed found that chemotherapy in combination with immunotherapy can be synergistic, activating the immune response by killing tumor cells, releasing tumor cell antigens, modulating T cell functions, reconfiguring the immune microenvironment and reducing the number of immunosuppressive cells[14–17]. However, chemotherapy overdoses directly or indirectly kills immune cells, reducing the number of circulating lymphocytes and monocytes and hindering the immune response against the tumor[18]. Therefore, we conducted this clinical trial of short course (2 cycles) chemotherapy plus the PD-1 inhibitor sintilimab as first-line treatment for advanced lung squamous cell carcinoma to assess the efficacy and tolerability of the treatment. Several studies have also shown that dynamic changes in circulating tumor DNA (ctDNA) are predictive or have a prognostic value for a variety of tumors[19–21]. We also collected and analyzed peripheral blood (PB) ctDNA dynamics.

In this work we show that sintilimab in combination with a reduced short-course of nab-paclitaxel/platinum led to encouraging PFS and OS times as first-line therapy for advanced sq-NSCLC.

## Results

### Baseline characteristics of patients

From May 2020 to June 2022, informed consent was obtained from 48 patients, and 47 of them underwent at least one treatment. Among the 48 enrolled patients, two were excluded from the data analysis—one was recruited before the protocol modification, and the other did not meet the eligibility criteria. In total 44 patients had evaluable imaging data and 3 were excluded from the efficacy analysis (1 due to protocol deviation, 1 who withdrew consent and 1 who died of fungal pneumonia after 1 cycle of treatment) (Fig. 1). At the cut-off date (August 31, 2023), 2 (4.5%) patients were still receiving treatment, while 42 discontinued treatment, including 29 who had disease progression (PD), 4 with intolerable adverse events (AEs), 1 accidental death (not due to disease), 4 voluntary withdrawals, 1 lost to follow-up, and 3 who completed 2 years of treatment (Fig. 1).

The enrolled patients' median age was 65 years (range: 52–76). Most patients were male (45/46, 97.8%), smokers (37/46, 80.4%), had an Eastern Cooperative Oncology Group Performance Status (ECOG PS) score of 1 (32/46, 69.6%) and stage IV disease (35/46, 76.1%).

Thirty-three patients had evaluable programmed cell death ligand-1 (PD-L1) expression results and 30.4% patients had a score ≥1% (Table 1).

In the present clinical trial, 11/46 (23.9%) of the patients were at stage IIIB/IIIC, which was in the same range as similar previous NSCLC

studies (ORIENT-12: 21.8–24.7%[11], CAMEL-Sq: 28%[22], RATIONALE 307: 31.7–36.4%[23]). Before their inclusion, these patients underwent a comprehensive review by a multidisciplinary tumor board. Except for 1 patient (stage IIIB) who declined radiotherapy, the remaining 10 were found to be unsuitable candidates for curative surgery or chest radiotherapy. The specific reasons are listed in Supplementary Table 1.

### Primary and secondary endpoints analyses

**Clinical efficacy.** At the cut-off date (August 31, 2023), the median follow-up time was 24.2 months (range: 1.0–37.8 months) and the trial met its primary endpoint, with a median PFS of 11.4 months [95% confidence interval (CI): 6.7–18.1 months]. The estimated PFS rate was 42.9% (95% CI: 27.9–57.1%) at 12 months and 25.1% (95% CI: 11.7–40.9%) at 24 months (Fig. 2a).

For the secondary endpoints, the objective response rate (ORR) was 70.5% (95% CI: 54.8–83.2%) with a partial response (PR) in 31 (70.5%) patients without a complete response. The disease control rate (DCR) was 93.2% (95% CI: 81.3–98.6%) (Table 2). The median time to response (TTR) was 1.4 months (range: 1.3–4.1 months), and the median duration of response (DOR) was 13.6 months [95% CI: 7.0–not evaluable (NE)] (Table 2), with an estimated DOR rate of 51.6% (95% CI: 33.0–67.4%) at 12 months and 32.2% (95% CI: 14.9–50.8%) at 24 months (Table 2 and Fig. 2b). Up to the last follow-up date, 20/44 (45.5%) patients had died. The median OS was 27.2 months (95% CI: 20.2–NE), with an OS rate at 12 months of 79.1% (95% CI: 63.6–88.5%) and a 24-month OS rate of 53.3% (95% CI: 35.5–68.2%) (Fig. 2c). The best percentage changes from baseline for each patient are shown in Fig. 2d.

Up to the cut-off date on August 31 2023, 3 patients completed 2 years of planned immunotherapy as scheduled. PD was observed in 29 patients (65.9%), and among them, 23 initiated second-line treatments, of whom 14 received chemotherapy (12 with gemcitabine + platinum-based agents, and 2 with paclitaxel + platinum-based agents), 5 received immunotherapy combinations (3 combined with chemotherapy, 2 in combination with anlotinib), and 3 were treated solely with anlotinib, while 1 patient underwent radiotherapy for bone metastatic lesions. The OS for second-line treatment patients was 21.9 months (95% CI: 16.4–32.4 months) (Supplementary Fig. 1).

**Safety.** A total of 46 patients were included in the safety analysis, and of these 42 completed the planned 2 cycles of chemotherapy. The median treatment duration was 7.8 months (range: 0.7–25.5 months) and the median number of treatment cycles was 11 (range: 1–35 cycles). Forty-two patients (91.3%) experienced treatment-related AEs (TRAEs), the most common being anemia (32.6%), elevated α-hydroxybutyrate dehydrogenase (28.3%) and elevated alanine aminotransferase (26.1%). Three patients (6.5%) experienced grade 3 TRAEs including 1 decreased white blood cell count, 1 immune-associated pneumonia and 1 rash. The last two TRAEs lead to treatment discontinuation, which was considered to be related to immunotherapy, while there were 2 non-TRAEs that lead to treatment discontinuation, 1 due to a sudden cardiovascular event and 1 due to a long-term pulmonary fungal infection. Two patients (4.3%) had grade 4 TRAEs of myelosuppression and continued treatment. No TRAEs caused death (Table 3). Immune-related AEs (irAEs) of any grade were reported for 14 (30.4%) patients. The most common irAE was elevated alanine aminotransferase [6 (13.0%) patients], followed by hypothyroidism [5 (10.8%) patients], hyperthyroidism [4 (8.7%) patients] and rash [2 (4.3%) patients]. Grade 3 irAEs were observed in 2 patients, 1 with pneumonitis and the other with a rash. There were no grade 4 or 5 irAEs (Table 4).

### Predefined exploratory analyses

**Subgroup analyses.** After univariate analyses, we did not find any clinical characteristic factor that was related to PFS and OS (Supplementary Tables 2 and 3).

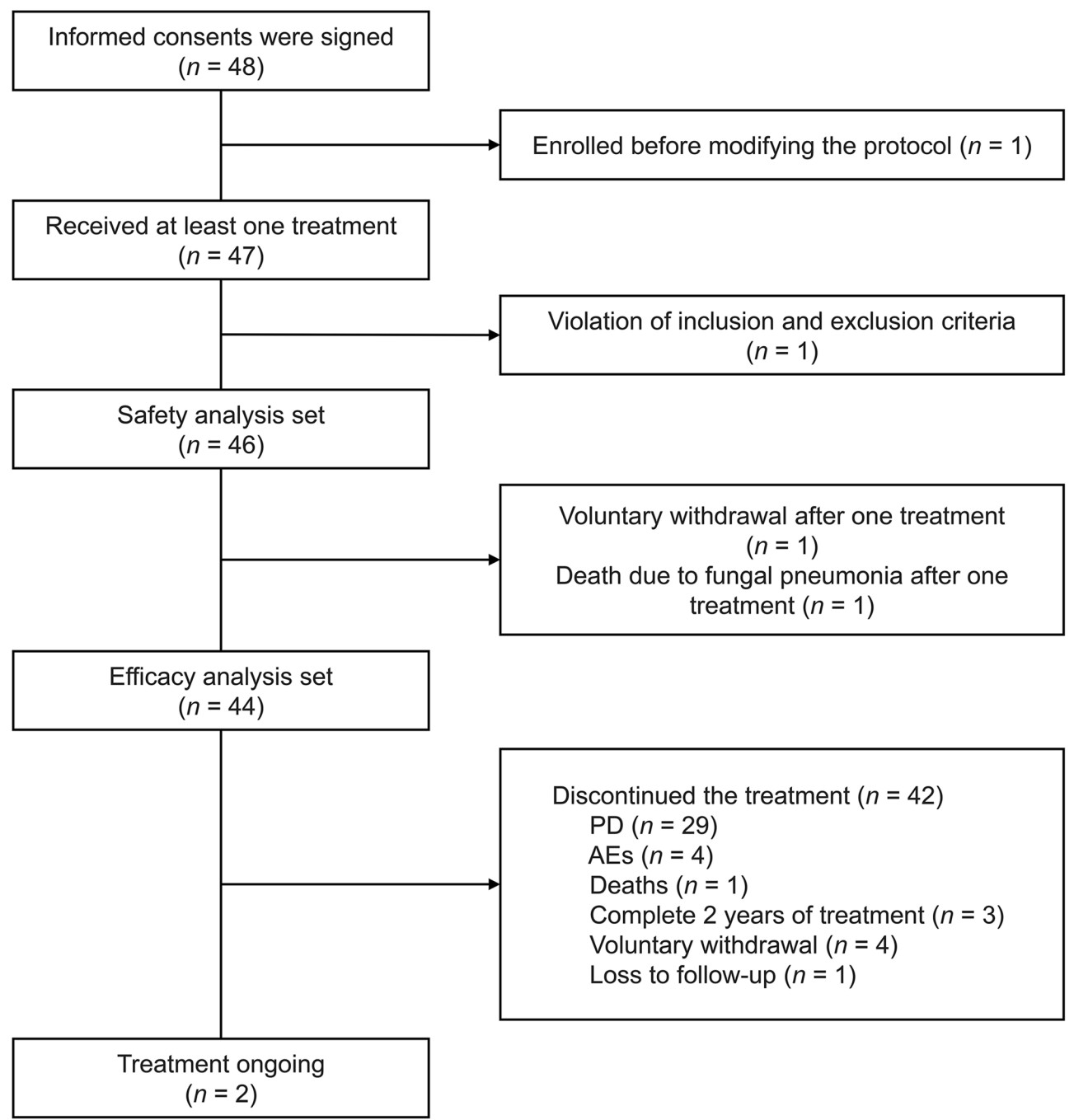

**Fig. 1 | Flow chart of the clinical trial (patient flow chart).** AEs adverse events, PD disease progression.

Furthermore, we analyzed the outcomes of subgroup populations by PD-L1 expression [tumor cell proportion score (TPS) <1%, $n = 19$, TPS = 1–49%, $n = 11$, TPS ≥50%, $n = 3$, not detectable (ND), $n = 11$]. Analysis showed that the ORR was 63.2% in TPS <1%, 81.8% in TPS = 1–49% and 66.7% in TPS ≥50%. There were no significant differences for PFS and OS among the three groups (Supplementary Table 4 and Supplementary Fig. 2). There was a significant difference in DOR among these four groups, which was longest in the TPS = 1–49% ($p < 0.001$) group.

**The relationship between ctDNA and treatment efficacy.** PB samples were collected from 26 patients and genomic abnormalities and ctDNA dynamics were evaluated at baseline (C0), after cycle 1 of treatment (C1) and after cycle 2 of treatment (C2). Four patients were ctDNA negative at baseline and 3 patients lacked sampling data at C1.

The rate of ctDNA positivity at baseline, C1 and C2 were 84.6% (22/26), 30.4% (7/23) and 15.4% (4/26), respectively. Mutations for each evaluable patient at baseline were profiled. The top altered genes of ctDNA-positive patients with mutant rates over 8% are listed in Fig. 3a. Survival analysis revealed that patients with breast cancer gene 2 (*BRCA2*), BMP/Retinoic Acid Inducible Neural Specific 3 (*BRINP3*), F-box/WD repeat-containing protein 7 (*FBXW7*), tyrosine-protein kinase KIT (*KIT*) or retinoblastoma 1 (*RB1*) abnormalities had shorter PFS times than those without abnormalities (Supplementary Fig. 3). Any one of these 5 genes (*BRCA2*, *BRINP3*, *FBXW7*, *KIT*, or *RB1*) mutations vs. wild-type cases led to median PFS times of 5.4 vs. 22.1 months, HR = 8.8; 95% CI: 2.4–32.0; $p < 0.001$) (Fig. 3b).

However, the efficacy of each biomarker during the baseline sample analysis by ROC curve showed that the predictive effectiveness

## Table 1 | Baseline characteristics of the patients

| Characteristic | Patients (N = 46) |
|---|---|
| **Age (years), median (range)** | 65 (52–76) |
| <65, n (%) | 19 (41.3) |
| ≥65, n (%) | 27 (58.7) |
| **Gender, n (%)** | |
| Male | 45 (97.8) |
| Female | 1 (2.2) |
| **Smoking status, n (%)** | |
| Current/Former | 37 (80.4) |
| Never | 9 (19.6) |
| **ECOG PS, n (%)** | |
| 0 | 14 (30.4) |
| 1 | 32 (69.6) |
| **Clinical stage, n (%)** | |
| IIIB/IIIC | 11 (23.9) |
| IV | 35 (76.1) |
| **Pathologic subtypes, n (%)** | |
| Squamous | 43 (93.5) |
| Mixed squamous NSCLC | 3 (6.5) |
| **PD-L1, n (%)** | |
| TPS <1% | 19 (41.3) |
| TPS = 1–49% | 11 (23.9) |
| TPS ≥50% | 3 (6.5) |
| ND | 13 (28.3) |

Source data are provided in the Source Data file.
*ECOG PS* Eastern Cooperative Oncology Group Performance Status, *ND* not detectable, *NSCLC* non-small cell lung cancer, *PD-L1* programmed cell death ligand-1, *TPS* tumor cell proportion score.

of the 18-month survival rate was significantly enhanced whenever a mutation occurs in any one of these five genes (*BRCA2, BRINP3, FBXW7, KIT*, or *RB1*), surpassing that of a single gene indicator (Supplementary Fig. 4).

We further focused on the relevance of ctDNA dynamics to clinical outcomes. Excluding 3 patients lacking blood samples at C1, we found that patients with ctDNA clearance either at C1 or C2 had better best responses than those with ctDNA uncleared (including baseline status $p = 0.034$; excluding baseline status $p = 0.046$, Fig. 3c, d). Analysis of ctDNA clearance and survival in all evaluable patients revealed that ctDNA clearance at C2 was associated with better PFS than those with uncleared (18.1 vs. 4.3 months, $p = 0.024$, HR = 0.2, 95% CI: 0.1–0.9). This result was consistent even when removing 4 patients with baseline ctDNA that was negative (17.3 vs. 4.3 months, $p = 0.035$, HR = 0.3, 95% CI: 0.1–1.0) (Fig. 3e, f).

In addition, the ctDNA changes of 23 patients that had ctDNA evaluations at both C1 and C2 were visualized (Fig. 3g). Twenty out of 23 patients were assessed as PR after sintilimab combined with chemotherapy and 19 underwent at least one ctDNA clearance at different time points. One PR patient showed a decrease in ctDNA at C2.

## Discussion

The present phase 2 trial was conducted on advanced sq-NSCLC patients who received a reduced number of 2 cycles of 260 mg/m² q3w nab-paclitaxel/carboplatin and sintilimab, followed by maintenance therapy with sintilimab. Previously two global trials, KEYNOTE-407 and IMpower131, confirmed the good efficacy of a full course of chemotherapy combined with ICI in untreated patients with advanced sq-NSCLC[9,10]. Subsequently two studies, Camel-sq and RATIONALE-307, also reported the clinical benefits of full dose chemotherapy plus

immunotherapy in China[22,23]. The median PFS of these trials ranged from 6.3 to 8.5 months (KEYNOTE-407, 8.0 months; IMpower131, A + CnP group 6.3 months; Camel-sq, 8.5 months; RATIONALE-307 Arm B, 7.6 months). ORR ranged from 49.7% to 72.5% (KEYNOTE-407, 57.9%; IMpower131 A + CnP group, 49.7%; Camel-sq, 64.8%; RATIONALE-307 group B, 74.8%)[9,22–24].

The present chemo-immunotherapy study with 2 cycles chemotherapy yielded results showing a median PFS time of 11.4 months (95% CI: 6.7–18.1) and a median OS time of 27.2 months (95% CI: 20.2–NE), while the ORR reached 70.5%, and the DCR was 93.2%, with a median DOR of 13.6 months (95% CI: 7.0–NE). In the KEYNOTE-407 study (chemo-immunotherapy, including 4–6 cycles chemotherapy), the median PFS time was 8 months (95% CI: 6.3–8.4), median OS was 17.1 months (95% CI: 14.4–19.9), ORR was 62.6% (95% CI: 56.6–68.3), and DOR was 8.8 months (1.3+ to 28.4+)[24]. In addition, in the RATIONALE-307 study (chemo-immunotherapy, including 4–6 cycles chemotherapy), the median PFS under tislelizumab plus chemotherapy [tislelizumab + paclitaxel and carboplatin (PC), tislelizumab + nab-PC] was 7.6 months (95% CI: 6.0–9.8) and 7.6 months (95% CI: 5.8–11.0), respectively. Notably, in the same study, for patients with stage IIIB disease, the median PFS extended to 9.8 and 11.0 months, ORR was 73% (95% CI: 63.6–80.3) for tislelizumab + PC and 75% (95% CI: 66.0–82.3) for tislelizumab + nab-PC, and DCR was 88% (95% CI: 80.2–92.8) for tislelizumab + PC and 91% (95% CI: 84.1–95.3) for tislelizumab + nab-PC[23]. Taken together, the anti-tumor effect of the present two-cycle treatment was comparable to the standard 4–6 cycles of chemotherapy combined with immunotherapy observed in other treatment regimens. The CheckMate 9LA, a double-blind phase 3 study of 2 cycles of chemotherapy in combination with dual immunotherapy reported the primary endpoint of median OS, which was significantly improved in the experimental group compared to the chemotherapy group (14.1 vs. 10.7 months; HR, 0.69; 95% CI: 0.55–0.87). Similarly, a significant improvement was also demonstrated for the median PFS (6.8 vs. 5.0 months) and ORR (37.7% vs. 25.1%)[12], while patients in the immunotherapy group experienced more severe TRAEs than those in the chemotherapy group (30% vs. 18%).

This clinical pilot trial of limited chemotherapy plus immune monotherapy for the treatment of advanced sq-NSCLC has revealed a PFS time of 11.4 months and the ORR of 70.5% was also significantly enhanced, with a median response time of 1.4 months, which was comparable to that achieved with a full course of chemotherapy plus immunotherapy[11], suggesting that short courses of chemotherapy may also improve the immune response at an early stage.

In addition, although the study included more elderly patients (≥65 years, 58.7%), the treatment was well tolerated. The incidence of grade 3 or 4 TRAEs was apparently lower (10.9%), and the proportion (4.3%, 2/46) of patients who experienced treatment discontinuation due to TRAEs was very rare (1 grade 3 of immune-associated pneumonia, 1 grade 3 of rash). No treatment-related deaths were reported. Thus, two cycles of chemotherapy can enhance the antitumor activity of immunotherapy without inhibiting immunogenicity and minimizes TRAEs.

The study included mostly males, which reflects the general sex disparity of sq-NSCLC in non-smoking Chinese[3]. In addition, previous research indicated a stronger correlation of smoking with squamous cell cancer than with adenocarcinoma[25], while smoking is extremely unpopular for Chinese women[26].

Nab-paclitaxel can be administered weekly and every 3-weeks. A clinical study that compared the efficacy and safety of the weekly regimen to the 3-week regimen in NSCLC reported that in the intent-to-treat (ITT) population, the weekly regimen demonstrated certain advantages in terms of ORR and safety. However, there were no significant differences in PFS and OS between the two administration regimens. Subgroup analyses revealed that squamous cell carcinoma patients had better PFS and OS when treated with the 3-week regimen,

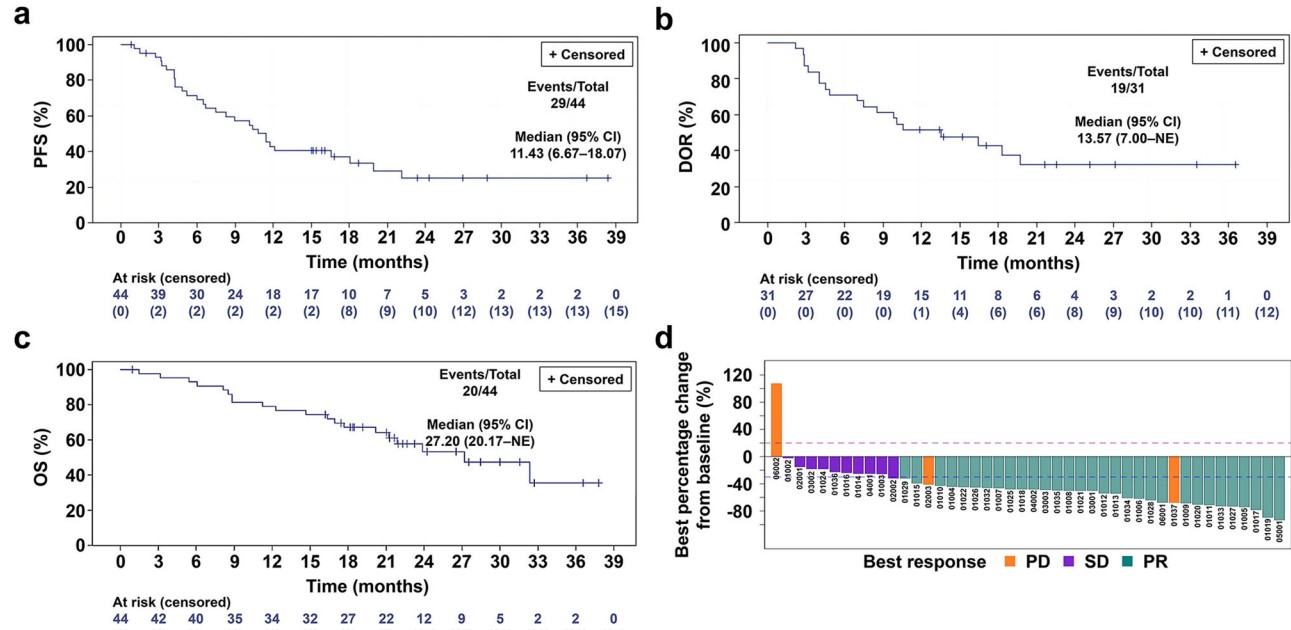

**Fig. 2 | Survival outcomes and tumor responses.** Progression-free survival (**a**), duration of response (**b**), overall survival (**c**) curves for the efficacy analysis set (*n* = 44), and (**d**) waterfall plot of best percentage change from baseline (*n* = 44). Patients 06001 and 02003 experienced PR only once, and were subsequently evaluated as PD. The optimal overall efficacy was evaluated as PD. For patient 01029, the efficacy evaluation was SD → SD → PR → PD, with only one optimal response of PR. the optimal overall efficacy evaluation was SD. CI confidence interval, DOR duration of response, NE not evaluable, OS overall survival, PD disease progression, PFS progression free survival, PR partial response, SD stable disease. Source data are provided in the Source Data file.

showing significant statistical differences. Among these, the 260 mg/m² q3w dosage exhibited better tolerability[27]. Additionally, in a study of neoadjuvant chemotherapy combined with immunotherapy for NSCLC conducted in China, nab-paclitaxel at a dosage of 260 mg/m² q3w was also chosen for sq-NSCLC, and the patients tolerated it well[28]. Considering this information, a 260 mg/m² q3w dosage was selected.

PB was collected from 26 patients for ctDNA dynamics analysis. Among them, the majority (22/26) showed ctDNA clearance at C2, resulting in a significantly higher ORR of 87.0%. Patients who tested negative for ctDNA at baseline (C0) and achieved clearance at C2 exhibited notably longer PFS compared to those with uncleared ctDNA (18.1 vs. 4.3 months, HR = 0.2, *p* = 0.024).

Several phase 3 clinical trials on sq-NSCLC have reported that no significant correlation was found between PD-L1 expression treatment efficacy or survival[22–24]. In the present trial, we also investigated the level of PD-L1 expression on tumor tissue. No correlation between PD-L1 expression level and PFS as well as OS was found, but the DOR was shortest for patients with the high PD-L1 expression, which may be related to the very small proportion

with high PD-L1 expression. In the context of advanced NSCLC treated with mono-immunotherapy, superior efficacy in patients with high PD-L1 expression might be observed compared to those with low or negative expression. However, in the context of combination therapy, such as immunotherapy with chemotherapy, the predictive value of PD-L1 expression appears to be reduced. For example, in the KEYNOTE-407 trial, it was noted that patients with PD-L1 expression (≥1%) might experience greater benefits in terms of PFS and OS. Nevertheless, the analysis of OS found no significant difference between patients with high PD-L1 expression (≥50%) and those with low PD-L1 expression (1–49%) (HR 0.79 vs. 0.59)[24], while in the RATIONALE-307 trial, no significant correlation was found between PD-L1 expression status and PFS. Additionally, patients with high PD-L1 expression (≥50%) exhibited higher HR values for PFS compared to those with low expression (1–49%) (tislelizumab plus PC: HR 0.50 vs 0.44; tislelizumab plus nab-PC: HR 0.43 vs. 0.31)[23]. Furthermore, in various randomized controlled trials for advanced squamous cell carcinoma of the lung, there was notable variability in the proportion of patients with high PD-L1 expression (ranging from 12.9% to 35%)[9,23], which may be linked to the higher tumor mutation burden in smoking-related squamous cell carcinoma patients.

In the present trial, we profiled the somatic mutations from baseline blood samples of the enrolled patients and explored potential predictive biomarkers for treatment with short-course chemotherapy combined with sintilimab as per our predefined exploratory endpoints. We discovered that patients harboring any of the *BRCA2*, *BRINP3*, *FBXW7*, *KIT* or *RB1* abnormalities had worse clinical outcomes, with median PFS becoming definitely shorter than those without abnormalities, findings also reported in previous studies[29–33]. Several studies have demonstrated a proven correlation between ctDNA clearance and clinical outcomes for a variety of tumor types[34–36], and PB ctDNA analyses of certain mutations have been proposed as surrogates for advanced lung cancer biopsy samples[37].

## Table 2 | Response data of the efficacy analysis set

| Best overall response | Efficacy analysis set (*n* = 44) |
|---|---|
| PR, *n* (%) | 31 (70.5) |
| SD, *n* (%) | 10 (22.7) |
| PD, *n* (%) | 3 (6.8) |
| ORR, *n* (%) [95% CI] | 31 (70.5) [54.8–83.2] |
| DCR, *n* (%) [95% CI] | 41 (93.2) [81.3–98.6] |
| Median TTR, months [95% CI] | 1.4 [1.3–4.1] |
| Median DOR, months [95% CI] | 13.6 [7.0–NE] |

Source data are provided in the Source Data file.
*DCR* disease control rate, *DOR* duration of response, *NE* not evaluable, *ORR* objective response rate, *PD* disease progression, *PR* partial response, *SD* stable disease, *TTR* time to response.

**Table 3 | Treatment-related adverse events**

| TRAEs | All grades *n* (%) | Grade 3–4 *n* (%) |
|---|---|---|
| Any grade | 42 (91.3) | 5 (10.9) |
| Grade 3 | | 3 (6.5) |
| Grade 4 | | 2 (4.3) |
| SAE | 5 (10.9) | 3 (6.5) |
| TRAEs leading to death | 0 | 0 |
| TRAEs leading to treatment discontinuation | 2 (4.3) | 2 (4.3) |
| ≥10% TRAEs | | |
| Anemia | 15 (32.6) | 0 |
| Elevated α-hydroxybutyrate dehydrogenase | 13 (28.3) | 0 |
| Elevated alanine aminotransferase | 12 (26.1) | 0 |
| Elevated aspartate aminotransferase | 10 (21.7) | 0 |
| Elevated blood lactate dehydrogenase | 10 (21.7) | 0 |
| Decreased serum albumin | 9 (19.6) | 0 |
| Elevated blood glucose | 9 (19.6) | 0 |
| Decreased platelet count | 7 (15.2) | 0 |
| Elevated γ-glutamyl transferase | 6 (13.0) | 0 |
| Decreased white blood cell count | 6 (13.0) | 1 (2.2) |
| Decreased thyrotropin stimulating hormone | 6 (13.0) | 0 |
| Elevated creatine kinase (MB form) | 6 (13.0) | 0 |
| Elevated creatine kinase | 6 (13.0) | 0 |
| Lacking in strength | 6 (13.0) | 0 |
| Elevated serum creatinine | 5 (10.9) | 0 |
| Elevated free thyroxine | 5 (10.9) | 0 |

Source data are provided in the Source Data file.
*SAE* serious adverse event, *TRAEs* treatment-related adverse events.

**Table 4 | Immune-related adverse events in this trial**

| irAEs | All grades *n* (%) | Grade 3 *n* (%) |
|---|---|---|
| Any irAE | 14 (30.4%) | 2 (4.3%) |
| Elevated alanine aminotransferase | 6 (13.0%) | 0 |
| Hypothyroidism | 5 (10.8%) | 0 |
| Hyperthyroidism | 4 (8.7%) | 0 |
| Rash | 2 (4.3%) | 1 (2.2%) |
| Fatigue | 2 (4.3%) | 0 |
| Elevated blood glucose | 2 (4.3%) | 0 |
| Elevated creatine kinase | 2 (4.3%) | 0 |
| Elevated creatine kinase (MB form) | 2 (4.3%) | 0 |
| Immune-mediated pneumonitis | 1 (2.2%) | 1 (2.2%) |
| Platelet count decreased | 1 (2.2%) | 0 |

Source data are provided in the Source Data file.
*irAE* immune-related adverse event.

Song et al. monitored ctDNA dynamics in a longitudinal cohort of advanced NSCLC patients and found that the ctDNA clearance at any timepoint during treatment was associated with a longer PFS[38]. Han et al. evaluated the predictive value of ctDNA clearance (C2) in second-line chemotherapy (6 weeks post-treatment), and found it to be an independent risk factor for PD or death[39]. The Camel-Sq trial revealed that the mean variant allele frequencies of variants at C2 had a strong association with ORR. Patients with baseline ctDNA negative and ctDNA clearance at C2 had significantly longer PFS (12.9 vs. 11.0 vs. 5.3 months; $p < 0.001$) and OS (NE vs. NE vs. 12.4 months; $p < 0.001$) than those with uncleared ctDNA[22]. A retrospective pooled analysis of RATIONALE-304 and RATIONALE-307 trials revealed that ctDNA levels from baseline to the first response were decreased ($p < 0.001$) and that patients with undetectable ctDNA at first remission had significantly longer median PFS and OS than those with detectable levels of ctDNA[40].

Our trial has demonstrated that the majority of patients with ctDNA clearance at C2 experienced better treatment effects, and patients who had negative ctDNA at baseline and achieved clearance at C2 exhibit significantly prolonged PFS. These results suggest a clinical utility of dynamic ctDNA monitoring during treatments with ICI plus chemotherapy, but a predictive impact should be further evaluated with trials that include a comparator arm.

Unfortunately, 4/44 patients died suddenly from COVID-19 during the opening period of the trial (October–December, 2022). The relationship between mortality and COVID-19 was therefore difficult to evaluate unequivocally.

In addition, the small sample size could only give a hint on the relationship between favorable outcomes on PFS with the combination of a reduced administration of conventional chemotherapy with a PD-1 inhibitor, which has been previously been reported for similar first line sq-NSCLC therapies with extended chemotherapy cycles[11,41], but beside enhancing the small sample size, further validation through experimental means are necessary to confirm the preliminary sequencing data findings. However, the findings in the present trial were encouraging, confirming not only clinical efficacy, but also the potential to achieve an extended period of disease control. Considering the single-arm trial design, and treatment discontinuation due to COVID-19, further clinical research with a larger sample size is needed to confirm definitively our findings, and moreover more tissue staining should be performed when tissue samples are available to identify further *BRCA2, BRINP3, FBXW7, KIT* and *RB1* mutations in tumor tissues.

In conclusion, sintilimab in combination with a short-course of nab-paclitaxel/platinum had an acceptable ORR and encouraging PFS and OS as first-line therapy for advanced sq-NSCLC. The evaluation of ctDNA dynamics during treatment revealed a correlation between ctDNA clearance and outcomes during the adjusted chemotherapy treatment combined with immunotherapy.

## Methods
### Patients
Eligible patients were: 18–75 years old; had histologically or cytologically confirmed advanced or metastatic sq-NSCLC without sensitizing *EGFR/ALK/ROS1* alterations; were at disease stage IIIB–IV; were unresectable or inappropriate for concurrent or sequential chemoradiotherapy [according to the American Joint Committee on Cancer (AJCC) 8th edition]; had not received prior systemic disease treatment; ECOG PS scores of 0 or 1; and had at least one measurable lesion according to the Response Evaluation Criteria In Solid Tumors (RECIST) 1.1 criteria.

Key exclusion criteria were: sq-NSCLC patients with sensitizing *EGFR/ALK/ROS1* alterations (test not mandatory); symptoms of central nervous system metastases; use of immunosuppressive drugs within 4 weeks before the trial treatment; with known or suspicious autoimmune diseases (congenital or acquired); and interstitial pneumonia. More detailed inclusion and exclusion criteria as well as any changes in enrolled patients before and after the protocol amendment are listed in Supplementary Methods.

The protocols were conducted in accordance with the Declaration of Helsinki and Good Clinical Practice guidelines[42] and approved by each participating institutions ethics review board of 6 study sites (Supplementary Table 5). The enrolment was from May 08, 2020 (first patient) to April 25, 2022 (last patient) and all patients signed written informed consent before participating in the trial and all enrolled

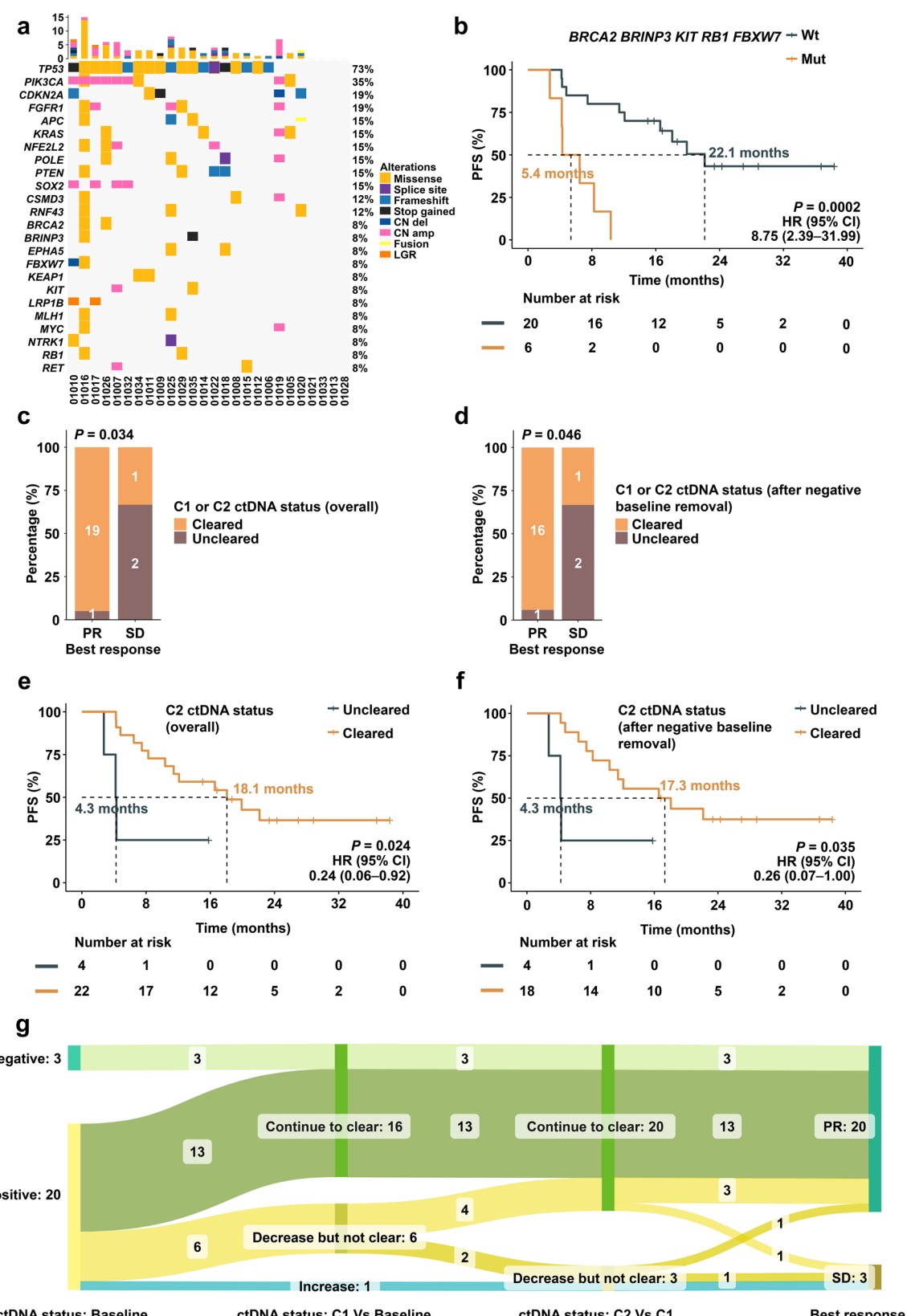

patients provided consent for the publication of potentially identifying clinical information.

## Trial design

This was a prospective, multi-center, open-label, single-arm, phase 2 clinical trial registered with the Chinese Clinical Trial Registry (link:

https://www.chictr.org.cn/, number: ChiCTR1900021726, date: 07/03/2019). It was originally designed and registered as version 1.0 on December 26, 2018 with the protocol of a randomized controlled phase 2 clinical study of short-course chemotherapy of paclitaxel/platinum with sintilimab versus paclitaxel/platinum as first-line treatment for locally advanced or metastatic sq-NSCLC in Chinese patients.

**Fig. 3 | Exploratory biomarker analyses. a** The top-ranked mutated genes in the studied cohort with colors representing different types of mutations (*n* = 26). **b** Kaplan−Meier survival curves depicting the PFS of patients carrying any of the *BRAC2, BRINP2, FBXW7, KIT, RB1* mutations or wild type. **c** The best response of patients with ctDNA clearance at C1 or C2 and those without ctDNA clearance including baseline clearance patients. **d** The best response of patients with ctDNA clearance at C1 or C2 and those without ctDNA clearance, excluding baseline clearance patients. **e** Kaplan−Meier survival curves depicting the PFS times of patients with ctDNA clearance or with ctDNA uncleared at C2 including baseline clearance patients. **f** Kaplan−Meier survival curves depicting the PFS times of patients with or without ctDNA clearance at C2 excluding baseline clearance patients. **g** Correlation between ctDNA dynamics and the clinical response. **b**, **e**, **f** *p* values were calculated using the one-sided log-rank test at a significance level of 0.05. **c**−**d** Two-sided Fisher's exact test. *BRCA2* breast cancer gene 2, *BRINP3* BMP/Retinoic Acid Inducible Neural Specific 3, CI confidence interval, ctDNA circulating tumor DNA, *FBXW7* F-box/WD repeat-containing protein 7, HR hazard ratio, *KIT* tyrosine-protein kinase KIT, Mut mutation, PFS progression-free survival, *RB1* retinoblastoma 1, Wt wild type. Source data are provided in the Source Data file.

PD-1 inhibitors in combination with chemotherapy became the standard treatment for advanced sq-NSCLC in 2019, so chemotherapy alone was no longer an acceptable therapy. Thus, we modified the protocol (V2.1, April 26, 2020) to sintilimab combined with two cycles (short-course) nab-paclitaxel/platinum as first-line therapy for locally advanced or metastatic sq-NSCLC in a phase 2, single-arm clinical study, here reported.

Eligible patients received sintilimab (100 mg/10 ml, 200 mg) with albumin paclitaxel (260 mg/m²) plus carboplatin (AUC = 5) by intravenous infusion every 3 weeks for 2 cycles, followed by maintenance therapy with sintilimab until PD, intolerable toxicity, death or up to 2 years. In the case of discontinuation of sintilimab due to AEs, administration of sintilimab could be interrupted for up to 12 weeks. Dose reductions were permitted for chemotherapy, but not for sintilimab.

Blood was obtained within 1 h prior to the administration of the initial treatment (C0), after cycle 1 of treatment (C1), after cycle 2 (C2) and at the time of PD. Serum was separated and stored at −80 °C for subsequent testing.

### Endpoints and assessment

The primary endpoint was PFS, defined as the time from the date of a patient signing informed consent to the date of PD or death. Secondary endpoints included: the ORR; DCR; DOR; OS (time from receipt of signed informed content to death or the last follow-up); and safety. Tumor responses were assessed by the investigator according to the RECIST version 1.1[43] every 6 weeks in the first year and every 12 weeks thereafter until PD. Safety outcomes were measured by the prevalence of TRAEs, which were graded based on NCI Common Terminology Criteria for Adverse Events (CTCAE) version 5.0. Changes in vital signs, physical examination results and laboratory tests (such as hematology, clinical biochemistry, urine analysis) before, during and after treatment were carefully recorded and assessed. Exploratory endpoints were predefined to evaluate the relationship between PD-L1 expression in tumor tissue and treatment efficacy, and to assess the relationship between biomarkers in PB and treatment efficacy, including but not limited to the dynamic changes in ctDNA overall mutational burden and efficacy.

PD-L1: Archival or freshly paraffin-embedded tumor tissues were first deparaffinized by a routine histological procedure. The staining followed the instructions supplied with the PD-L1 the immunohistochemistry 22C3 pharmDx assay kit (Alilent Technologies, Dako, CA, USA) and were carried out at the central laboratory of the Affiliated Cancer Hospital of Zhengzhou University. Specimens were tested on the platform Dako ASL48. TPS was determined to measure the expression of PD-L1.

Cell-free DNA (cfDNA) and ctDNA: According to the protocol, 10 ml samples of PB were collected in EDTA tubes at each time point. Plasma and PB lymphocytes were separated within 2 h of collection and aliquoted for storage at −80 °C. Briefly, circulating cfDNA was extracted from a 5 ml of plasma sample using a QIAamp Circulating Nucleic Acid kit (Qiagen, Hilden, Germany). Fragments between 200 and 400 bp from the sheared and extracted cfDNA were purified (Agencourt AMPure XP Kit, Beckman Coulter, CA, USA), hybridized with capture probe baits, selected, amplified, and then subjected to targeted capture using a commercial panel consisting of 520 genes (OncoScreen Plus), spanning 1.64 megabases of the human genome. The size and quality of these fragments were assessed using a Bioanalyzer 2100 (Agilent Technologies, CA, USA). The ctDNA of all samples were processed using next-generation sequencing as previously described[37,44]. The indexed samples were sequenced on Nextseq 500 (Illumina, Inc., CA, USA) with paired-end reads and an average sequencing depth of ×10,000 for plasma samples. Sequence data analysis was conducted with a bioinformatics pipeline as previously described[45]. The maximum allele frequency of somatic mutations in plasma was used to determine the changes of ctDNA[46].

### Statistical analysis

In this trial PFS was used as the primary endpoint and the objective was to investigate whether a short course of chemotherapy combined with immunotherapy could prolong PFS in sq-NSCLC patients. According to published data, the median PFS for 4–6 cycles of the historic control chemotherapy with a paclitaxel/cisplatin (or carboplatin) regimen was 4.4 months[22,47] and it has been hypothesized that the addition of maintenance therapy with sintilimab after 2 cycles of sintilimab plus paclitaxel (nab-paclitaxel)/cisplatin (or carboplatin) would yield a 45% PFS improvement, resulting in a duration of 6.4 months as compared to the published median PFS time. To assess this hypothesis, the HR of λ1/λ0 was -0.69 by using a log rank test with a single bias type I error of 0.10, followed by a power of 80%. With an expected enrolment time of at least 12 months, and a follow-up time of 12 months, a minimum of 41 patients were needed to be enrolled to observe 34 events. Considering a 20% dropout rate, a total of 50 patients were enrolled in this study.

All analyses were based on the ITT principle. Patients in the efficacy analysis set received at least 1 dose of sintilimab and/or chemotherapy and had at least 1 tumor response evaluation. The safety analysis set refers to patients who received at least one dose of sintilimab and/or chemotherapy.

The data collection was conducted using 91Trial software version 1.0 (EDC system, Shanghai Ashermed Healthcare Communications Co., Ltd, Shanghai, China). All statistical analyses were performed using R software (R version 4.1.2) and SAS 9.4 (SAS Institute Inc., Cary, NC). The significance with categorical variables was evaluated using Fisher's exact test. The significance with continuous variable variables was evaluated using the Wilcoxon rank sum test. Survival curves were generated using the Kaplan−Meier method, and median survival time along with 95% CIs were estimated using the log transformation method. The log-rank test was employed to compare PFS, OS and DOR among different PD-L1 expression levels (<1%, 1–49%, ≥50%, ND). Considering the small sample size, the risk factors for PFS and OS were explored using univariate Cox regression analysis. The Sankey figure was plotted using an online visualization tool (https://sankeymatic.com/build/). The genomic alteration figure was plotted using ComplexHeatmap package[48]. The ROC curve was plotted using pROC package[49]. A *p* value < 0.05 was considered to be a statistically significant finding.

## Reporting summary

Further information on research design is available in the Nature Portfolio Reporting Summary linked to this article.

## Data availability

The minimum datasets necessary to interpret this research have been provided within the article, Supplementary Information and Source Data file, where applicable. ctDNA sequencing data are available at the GSA-Human dataset HRA006484 under restricted access for ethical and privacy concerns. Access can be granted through the Data Access Committees (DAC) of the GSA-Human database (2024BAT00029), with an estimated response time of about five working days for access requests. Upon approval, the data will be accessible for a duration of 6 months. Applicants may also directly contact the corresponding author, H.W. The original study protocol was written in Chinese, but a translated simplified version is available in the Supplementary Information file. Source data are provided with this paper.

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

## Acknowledgements

This study was supported by Innovent Biologics, Inc. The funding body had no role in the conceptualization, design, data collection, analysis, decision to publish, or preparation of the manuscript. We thank the patients and their families for making this trial possible, and the investigators and clinical research teams who participated. We are also grateful to Ying Sun, Wenbo Dong, Shuo Shi, Bing Li, and Lan Su from Burning Rock Biotech for technical assistance. We thank Shanghai Ashermed Healthcare Communications Ltd. for their data management, statistical analysis service and technical assistance. Medical writing support was provided by Shanghai BIOMED Science and Technology Co., Ltd., and funded by Innovent Biologics, Inc.

## Author contributions

M.N. Zhang contributed conceptualization, data curation, statistical analysis, writing—original draft and writing—review and editing; H.J. Wang contributed to conceptualization, analysis and interpretation, statistical analysis, writing—original draft and writing—review and editing; G.W. Zhang, Y.Y. Niu, G.F. Zhang, Y.H. Ji, X.T. Yan, X.J. Zhang, Q.CH. Wang, X.H. Jing, J.SH. Wang, and Z.Y. Ma contributed to data curation and writing—review and editing.

## Competing interests

The authors declare no competing interests.
