## [Peer Review File · Nature Communications]

Sintilimab with two cycles of chemotherapy for the treatment of advanced squamous non-small cell lung cancer: a phase 2 clinical trialREVIEWER COMMENTS

Reviewer #1 (Remarks to the Author): with expertise in lung cancer, therapy

In this manuscript, Zhang and colleagues describe a Phase 2 trial concerning Sintilimab combined with chemotherapy in 2 cycles followed by Sintilimab maintenance treatment for advanced stage lung squamous carcinoma. The authors show PFS and OS were 10.7 and 21.9 months, respectively. The data was extremely superior than previous results which was demonstrated by ORIENT-12. This prospective phase III clinical trial (ORIENT-12, NCT03629925) demonstrated that the treatment of Sintilimab combined with chemotherapy at least 4 cycles followed by Sintilimab maintenance could provide a median PFS of 5.1 months. Besides, the prospective phase III clinical trial (KEYNOTE-189, NCT02578680) demonstrated that the treatment of Pembrolizumab combined with chemotherapy could provide a median PFS of 8.8 months.

The authors also perform NGS and cfDNA in a subset of patients, which attempted to contribute the promising biomarkers of survival and therapy. This manuscript seems to be an interesting trial, but I think it is unreasonable to evaluate the effect of first-line treatment with 44 people in stage III-IV, where these various diseases are consisted.

1. Zhou C, et al. Sintilimab Plus Platinum and Gemcitabine as First-Line Treatment for Advanced or Metastatic Squamous NSCLC: Results From a Randomized, Double-Blind, Phase 3 Trial (ORIENT-12). *J Thorac Oncol.* 2021 Sep;16(9):1501-1511.
2. Gandhi L, et al. Pembrolizumab plus Chemotherapy in Metastatic Non-Small-Cell Lung Cancer. *N Engl J Med.* 2018 May 31;378(22):2078-2092.

Major issues

1. As a phase 2 clinical trial, the purpose of this research assesses the efficacy of short-course chemotherapeutic regimen for sq-NSCLC in advanced stage. However, the data of prognosis in this single-arm trial could not validate the purpose, because this study lacked a comparison with standard treatment. What is more, considering the different stage and PD-L1 expression level of tumor probably significantly influence the therapeutic efficacy, therefore, the efficacy of short-course therapy cannot be confirmed by comparing with data from both prospective clinical trials or real-world studies were published previously.

2. As in previous study, Pembrolizumab (KeyNote-042) can be used as a monotherapy for patients with high PD-L1 expression and has acceptable therapeutic efficacy. However, in this study, the authors attempted to evaluate the efficacy of short-term chemotherapy for NSCLC, and the proportion of patients with high PD-L1 expression should be discussed, rather than only dividing the PD-L1 expression level of patient into expression and non-expression subgroups according to the existing statistical data.
3. This samples size is too small to identify the efficacy of this strategy for sq-NSCLC in stage IIIB-IV, which were composed of multiple stage. The treatment strategies of stage IIIB-C would not be unified and their prognosis is heterogeneous. Considering that the proportion of stage III patients included in this study exceeds 20%, please carefully describe the reasons why these patients did not undergo radical surgery, neoadjuvant treatment followed by surgery, or PACIFIC regimens. For example, a portion of patients in stage IIIB who received induction immuno-chemotherapy may undergo radical resection to improve survival.
4. According to my understanding, the authors may believe that these biomarkers screened by NGS have the same prognostic efficacy, but I strongly recommend that the authors should profoundly discuss the role and efficacy of each biomarker during the baseline sample analysis by ROC curve. After that, tissue staining should be performed where available to determine whether BRCA2, BRINP3, FBXW7, KIT and RB1 are expressed in tumor cells.
5. Considering authors attempted to establish the relationship between the therapeutic efficacy and the dynamic changes of ctDNA during treatment, whether the relationship between the tumor WES and ctDNA should be evaluated.

Minor issues

1. The expression on page 6, line 136 is difficult to understand, please describe it again. Why baseline samples including C1 and C2 samples.
2. The median follow-up time of this study was 15.2 months, and it is inappropriate to discuss the 2-year survival rate in the section of clinical efficacy (Page 5, Line 98).
3. In Supplementary Tables 1 and 2, there are obvious errors in the statistical methods, and in the case of insufficient sample size (n=44), it is not possible to include all variables into the multivariable Cox regression model.
4. In Supplementary Fig. 1, please demonstrated the data of patients whose PD-L1

expression undetectable.

5. Please describe the primary endpoint in the Abstract and whether the study has met this point.
6. Please describe the secondary endpoints of the study in the Abstract, as well as the exploratory endpoints mentioned in the study protocol.
7. For endpoints which was listed in protocol but not mentioned in the manuscript, please explain the reasons.
8. Considering the OS was discussed by authors, the second-line treatment should also be shown in the article.
9. The last follow-up time is January 2023, please update the OS and PFS. Timely prognostic information is the most important endpoint in clinical trial report.
10. In the Abstract or in the Main Text, the contents that first appear should not be expressed as Abbreviation.

Reviewer #2 (Remarks to the Author): with expertise in lung cancer, therapy

Zhang and colleagues provide results of a single-arm phase 2 multi-center trial conducted in China. 47 Patients received two cycles of carboplatinum, nab-paclitaxel and sintilimab for stage III/IV squamous NSCLC, followed by ICI maintenance. The design of the trial is clearly explained, methodology is sound, results are clearly reported and the conclusions drawn are comprehensible. The trial met its primary endpoint (PFS) and demonstrated encouraging efficacy (ORR, DoR) and survival results (PFS, OS). Furthermore, responses were correlated with biomarkers (blood- and tissue-based) and ct-DNA clearance.

The authors should be congratulated for having conducted this trial. ORR, PFS and OS compare very favorably to what has been reported from comparable trials (KN407, 9LA).

There are some points that make the comparability to established regimens difficult. First, a relatively high share of patients with stage III disease was included (25%; 9LA/KN407: stage IV only, Empower-Lung 3 14% (sq only not reported)), and overall, the number of patients included is small. Second, the estimation of the real effect size is difficult due to the single-arm design. Third, RECIST evaluation was done by the local investigators and was not confirmed centrally. And forth, follow-up time is quite short (15 months; only 21 PFS and 18 OS events). Concerning baseline characteristics there is some overweight of male patients,

and the share of patients with undocumented PD-L1 expression level is relatively high (>50%). The nab-paclitaxel dose chosen (260 mg/m² q3w) does not represent an established dose regimen for NSCLC (at least in Europe or the U.S.) and should be commented. Whereas TRAE appear to be low, irAE are currently not reported, which I would recommend. Results for biomarkers confirm the current estimation but do not add substantially new information (ct-DNA clearance is associated with favorable outcome; correlation of gene analysis with outcome is prognostic, but as the trial lacks a comparator arm no conclusions can be drawn with regard to a predictive impact).

To conclude, the present investigation adds evidence for palliative first-line treatment of sq NSCLC and suggests that the number of initial cycles of chemotherapy might be limited to two. But we can not categorize the results with double checkpoint inhibition on the one side (CM9LA) and “classical” chemoimmunotherapy on the other side (KN407).

Reviewer #3 (Remarks to the Author): with expertise in biostatistics, clinical trial study design

This is an open-label single-arm phase 2 clinical trial of sintilimab, a PD-1 inhibitor, with two cycles of chemo for treatment-naïve advanced squamous NSCLC. The primary endpoint is PFS, with second endpoints including ORR, TTR, DOR, OS, and safety. The manuscript is well written with study design, methods and results.

1. The gender is unbalanced (only 1 female included). It's curious why this happened.
2. The Sintilimab is a PD-1 inhibitor, but only half patients tested for PD-L1 and analysis was stratified by TPS<1% VS >1%, why the other half is ND (not detectable)? Why PD-L1 test is not used in screening for everyone?
3. Line 114 said Cox univariate and multivariate analyses, but Supp Table 1 and 2 only shows multivariate analysis results, please correct.
4. Figure 3c and 3d did not specify which test was used to compare the response between ctDNA status, is the p-value from Fisher's exact test? Please describe in methods.

Reviewer #1 (Remarks to the Author): with expertise in lung cancer, therapy

In this manuscript, Zhang and colleagues describe a Phase 2 trial concerning Sintilimab combined with chemotherapy in 2 cycles followed by Sintilimab maintenance treatment for advanced stage lung squamous carcinoma. The authors show PFS and OS were 10.7 and 21.9 months, respectively. The data was extremely superior than previous results which was demonstrated by ORIENT-12. This prospective phase III clinical trial (ORIENT-12, NCT03629925) demonstrated that the treatment of Sintilimab combined with chemotherapy at least 4 cycles followed by Sintilimab maintenance could provide a median PFS of 5.1 months. Besides, the prospective phase III clinical trial (KEYNOTE-189, NCT02578680) demonstrated that the treatment of Pembrolizumab combined with chemotherapy could provide a median PFS of 8.8 months. The authors also perform NGS and cfDNA in a subset of patients, which attempted to contribute the promising biomarkers of survival and therapy. This manuscript seems to be an interesting trial, but I think it is unreasonable to evaluate the effect of first-line treatment with 44 people in stage III-IV, where these various diseases are consisted.

Reply: We agree and have addressed this concern in the limitations section of the revised manuscript and added 2 extra relevant references. In addition, the small sample size could only give a hint on the relationship between favorable outcomes on PFS with the combination of a reduced administration of conventional chemotherapy with a PD-1 inhibitor, which has been previously been reported for similar first-line sq-NSCLC therapies with extended chemotherapy cycles (Gandhi et al., 2018; Zhou et al., 2021).

- Zhou, C. et al. Sintilimab Plus Platinum and Gemcitabine as First-Line Treatment for Advanced or Metastatic Squamous NSCLC: Results From a Randomized, Double-Blind, Phase 3 Trial (ORIENT-12). *J. Thorac. Oncol.* 16, 1501-1511 (2021).
- Gandhi, L. et al. Pembrolizumab plus Chemotherapy in Metastatic Non-Small-Cell Lung Cancer. *N. Engl. J. Med.* 378, 2078-2092 (2018).

Major issues

1. As a phase 2 clinical trial, the purpose of this research assesses the efficacy of short-course chemotherapeutic regimen for sq-NSCLC in advanced stage. However, the data of prognosis in this single-arm trial could not validate the purpose, because this study lacked a comparison with standard treatment. What is more, considering the different stage and PD-L1 expression level of tumor probably significantly influence the therapeutic efficacy, therefore, the efficacy of short-course therapy cannot be confirmed by comparing with data from both prospective clinical trials or real-world studies were published previously.

Reply: Initially we designated solely chemotherapy as the control group when devising the trial. However, it is worth noting that in 2019, the landscape of cancer treatment in China evolved, with chemotherapy now being complemented by immunotherapy as the new standard treatment regime. As a result, standalone chemotherapy is no longer considered the optimal choice of treatment for patients. We therefore adjusted our trial protocol, as documented in the attached file labeled Protocol No CIBI308Y014. Due to considerations related to securing trial funding and the practical limitations on the number of eligible patients, we ultimately proceeded with a single-arm trial design.

We have mentioned these points in the revised Trial Design section of the manuscript. It is important to realize that it is our intention to conduct a randomized controlled phase 3 clinical trial in the near future. When the time comes, we will gladly welcome and incorporate your valuable suggestions, particularly the inclusion of an appropriate control group.

2. As in previous study, Pembrolizumab (KeyNote-042) can be used as a monotherapy for patients with high PD-L1 expression and has acceptable therapeutic efficacy. However, in this study, the authors attempted to evaluate the efficacy of short-term chemotherapy for NSCLC, and the proportion of patients with high PD-L1 expression should be discussed, rather than only dividing the PD-L1 expression level of patient into expression and non-expression subgroups according to the existing statistical data.

Reply: We appreciate your concern and in response, we have included the expression of PD-L1 in the analyses. Currently, we have acquired PD-L1 expression data for 33 patients. Of these, only 3 displayed high PD-L1 expression (PD-L1 \geq 50%), while 11 exhibited moderate PD-L1 expression (PD-L1 1-49%) and 19 low PD-L1 expression (PD-L1 $<$ 1%). There were no significant differences in PFS or OS times in patients with different PD-L1 expression levels, but significant differences of DORs were found between the groups.

We have added this information to the revised manuscript.

Supplementary Table 4 | Comparison of ORR, PFS, OS and DOR between the subgroup populations with different PD-L1 expressions ($n = 44$)

TPS $<$ 1%	TPS = 1–49%	TPS \geq 50%	P -value	PD-L1 ND	P -value
------------	-------------	----------------	-----------------	----------	-----------------

	n = 19	n = 11	n = 3	(< 1% vs. 1–49% vs. ≥ 50%)	n = 11	(for 4 groups)
ORR (%)	63.2	81.8	66.7	0.634	72.7	0.770
PFS (m), median (95% CI)	5.9 (4.2–21.9)	11.3 (3.2–NE)	5.1 (4.2–NE)	0.526	11.6 (6.4–16.3)	0.696
OS (m), median (95% CI)	21.9 (16.4–NE)	NE (8.8–NE)	NE (8.1–NE)	0.733	27.2 (8.5–NE)	0.882
DOR (m), median (95% CI)	16.4 (2.8–NE)	NE (4.0–NE)	2.5 (2.2–2.8)	0.010	10.6 (2.0–NE)	0.019

CI confidence interval, *DOR* duration of response, *HR* hazard ratio, *m* month, *ND* not detectable, *NE* not evaluable, *ORR* objective response rate, *OS* overall survival, *PFS* progression free survival, *TPS* tumor cell proportion score.

3. This samples size is too small to identify the efficacy of this strategy for sq-NSCLC in stage IIIB-IV, which were composed of multiple stage. The treatment strategies of stage IIIB-C would not be unified and their prognosis is heterogeneous. Considering that the proportion of stage III patients included in this trial exceeds 20%, please carefully describe the reasons why these patients did not undergo radical surgery, neoadjuvant treatment

followed by surgery, or PACIFIC regimens. For example, a portion of patients in stage IIIB who received induction immuno-chemotherapy may undergo radical resection to improve survival.

Reply: In present trial, we enrolled 11 patients at stage III. Before their inclusion, these patients underwent a comprehensive review by a multidisciplinary tumor board. Except for 1 patient (stage IIIB) who declined radiotherapy, the remaining 10 patients were found to be unsuitable candidates for curative surgery or chest radiotherapy.

The specific reasons for their ineligibility, which encompassed various conditions, have been added to the revised manuscript as Supplementary Table 1.

Subsequently, we also made a comparison with a previously conducted phase 3 clinical trial involving chemotherapy combined with immunotherapy for NSCLC, which included a certain proportion of stage III patients similar to those in the present trial (ORIENT-12: 21.8-24.7%; Camel-Sq: 28%; and RATIONALE 307: 31.7-36.4%). Therefore, our inclusion of 11 patients aligns with the proportions observed in previous studies.

Supplementary Table 1 | Reasons why patients did not undergo radical surgery, neoadjuvant treatment followed by surgery or PACIFIC regimens

Number	Clinical stage	Reason
01002	IIIC	Extensive cancer-associated lymphangitis
01009	IIIB	Huge tumor mass, and the boundary between the mass and the oesophagus is unclear

01011	IIIC	High tumor burden, extensive lymph node involvement, low lung function, high risk for radiotherapy
01013	IIIC	Multiple nodules in different lung lobes on the same side
01022	IIIC	Bilateral enlargement of hilar lymph nodes, unsuitable for radiotherapy
01025	IIIC	The patient declined radiotherapy
01029	IIIB	Bilateral enlargement of hilar lymph nodes, unsuitable for radiotherapy
01030	IIIB	Satellite lesions scattering in the same lung lobe, unsuitable for radiotherapy
01034	IIIC	An extensive tumor (10.5 cm) involving left main bronchus and causing complete lung collapse
03003	IIIB	A large cystic lesion containing an air-fluid level
06001	IIIB	Recurrent left lung cancer invading the left and right main bronchus, high risk for radiotherapy

4. According to my understanding, the authors may believe that these biomarkers screened by NGS have the same prognostic efficacy, but I strongly recommend that the authors should profoundly discuss the role and efficacy of each biomarker during the baseline sample analysis by ROC curve. After that, tissue staining should be performed where available to determine whether BRCA2, BRINP3, FBXW7, KIT and RB1 are expressed in tumor cells.

Reply: Thank you for your helpful comments. We first explored the association between PFS and gene mutations, and identified that mutations in *BRCA2*, *BRINP3*, *FBXW7*, *KIT* or *RB1* showed a similar association with poorer PFS ($P < 0.05$, Supplementary Fig. 3).

However, due to the relatively small sample size of patients with mutated genes, conducting additional stratified analyses may introduce a certain degree of bias. Therefore, we chose to examine the likelihood of any occurrence among 5 genes (*BRCA2*, *BRINP3*, *FBXW7*, *KIT* or *RB1*) and simultaneously assessed the predictive effects on survival rates using ROC curves for each individual gene (*BRCA2*, *BRINP3*, *FBXW7*, *KIT* or *RB1*). From Supplementary Fig. 4, it is evident that the predictive effectiveness of the 18-month survival rate is significantly enhanced whenever a mutation occurs in any one of these 5 genes (*BRCA2*, *BRINP3*, *FBXW7*, *KIT* or *RB1*), surpassing that of a single gene as an indicator.

In addition, we currently lack a sufficient quantity of tissue samples (some patients have died) for staining and verification in tumor cells, but we will conduct these tests in a further clinical trial in the very near future.

Since the number of patients with each mutation in the above genes was small, we further combined all mutations and observed a shorter PFS time than in those patients without any of these mutations (Fig. 3B).

Supplementary Fig. 3 | Survival analysis for patients with or without *BRCA2*, *BRINP3*, *FBXW7*, *KIT* or *RB1* abnormalities.

BRCA2 breast cancer gene 2, *BRINP3* BMP/Retinoic Acid Inducible Neural Specific 3, *CI* confidence interval, *ctDNA* circulating tumor DNA, *FBXW7* F-box/WD repeat-containing protein 7, *HR* hazard ratio, *KIT* tyrosine-protein kinase KIT, *Mut* mutation, *PFS* progression-free survival, *RB1* retinoblastoma 1, *Wt* wild type.

Supplementary Fig. 4 | ROC curve for the prognostic efficacy of predictive biomarkers.

5. Considering authors attempted to establish the relationship between the therapeutic efficacy and the dynamic changes of ctDNA during treatment, whether the relationship between the tumor WES and ctDNA should be evaluated.

Reply: Thank you for your helpful suggestion. Since all enrolled patients had locally advanced or advanced stages of disease without surgical intervention, it was difficult to obtain enough tissue samples for whole-exome sequencing (WES) analyses. However, the ctDNA sequencing method used in the present trial revealed a high concordance rate with tumor tissues in non-small cell lung cancer reported in a previous study (Mao et al., 2017). In the present trial, we used the maximum allele frequency of somatic mutations in plasma to monitor the ctDNA changes (Tang et al., 2019).

We have added this information to the Methods and Patients and Discussion sections of the revised manuscript.

- Mao, X. et al. Capture-Based Targeted Ultradeep Sequencing in Paired Tissue and Plasma Samples Demonstrates Differential Subclonal ctDNA-Releasing Capability in Advanced Lung Cancer. *J. Thorac. Oncol.* 12, 663-672 (2017).
- Tang, Y. et al. Maximum allele frequency observed in plasma: A potential indicator of liquid biopsy sensitivity. *Oncol. Lett.* 18, 2118-2124 (2019).

Minor issues

1. The expression on page 6, line 136 is difficult to understand, please describe it again.

Why baseline samples including C1 and C2 samples.

Reply: We have now made this information clearer.

2. The median follow-up time of this study was 15.2 months, and it is inappropriate to discuss the 2-year survival rate in the section of clinical efficacy (Page 5, Line 98).

Reply: Thank you for raising this question. We have updated the follow-up time to August 31, 2023; the median follow-up time was 24.2 months (ranging from 1.0 to 37.8 months).

Currently, 20 patients (45.5%) have experienced mortality, with a 12-month overall survival (OS) rate of 79.1% (95% CI: 63.6–88.5%) and a 24-month OS rate of 53.3% (95% CI: 35.5–68.2%).

We have added this new information to the revised manuscript.

3. In Supplementary Tables 1 and 2, there are obvious errors in the statistical methods, and in the case of insufficient sample size (n=44), it is not possible to include all variables into the multivariable Cox regression model.

Reply: Thank you for pointing out these errors. We have added univariable analysis to both Supplementary Tables 2 and 3.

4. In Supplementary Fig. 1, please demonstrated the data of patients whose PD-L1 expression undetectable.

Reply: As per your suggestion, we have added new patient data analyses including the ND data.

Supplementary Fig. 2 | Comparison of ORR, PFS, OS and DOR between the subgroup populations with different PD-L1 expressions (n = 44)

CI confidence interval, *DOR* duration of response, *HR* hazard ratio, *m* month, *ND* not detectable, *NE* not evaluable, *ORR* objective response rate, *OS* overall survival, *PFS* progression free survival.

5. Please describe the primary endpoint in the Abstract and whether the study has met this point.

Reply: The trial reached its primary endpoint of PFS, information which has been added to the revised Abstract.

6. Please describe the secondary endpoints of the study in the Abstract, as well as the exploratory endpoints mentioned in the study protocol.

Reply: The secondary endpoints included the objective response rate (ORR) (with partial response (PR) and complete response (CR)), disease control rate (DCR), time to response (TTR), duration of response (DOR), overall survival (OS) and safety indicators.

The exploratory endpoints included the relationship between biomarkers in tumor tissue and treatment efficacy, and evaluation of the association between ctDNA in peripheral blood and treatment efficacy.

7. For endpoints which was listed in protocol but not mentioned in the manuscript, please explain the reasons.

Reply: In this research, we performed an analysis and reported on the primary trial endpoint, PFS, as well as the secondary trial endpoints (ORR, DCR, TTR, DOR, OS,

and safety indicators) mentioned in the protocol. However, due to limitations, such as constrained funding and a scarcity of tissue samples, we were unable to collect the expected data for exploratory endpoints, except for the changes in blood ctDNA during the treatment process and its correlation with efficacy.

8. Considering the OS was discussed by authors, the second-line treatment should also be shown in the article.

Reply: We have added the following information to the revised Results section: Up to the cut-off date on August 31 2023, 3 patients completed two years of planned immunotherapy as scheduled. PD was observed in 27 patients (61.4%), and among them, 22 initiated second-line treatments, of whom 14 received chemotherapy (12 with gemcitabine + platinum-based agents, and 2 with paclitaxel + platinum-based agents), 5 received immunotherapy combinations (3 combined with chemotherapy, 2 in combination with anlotinib), and 2 were treated solely with anlotinib, while 1 patient underwent radiotherapy for bone metastatic lesions. The OS for second-line treatment patients was 21.2 months (95% CI: 14.7–32.4 months) (Supplementary Fig. 1).

Supplementary Fig. 1 | Kaplan-Meier curve of OS times for second line treatments.

CI confidence interval, *OS* overall survival.

9. The last follow-up time is January 2023, please update the OS and PFS. Timely prognostic information is the most important endpoint in clinical trial report.

Reply: Based on your recommendation, we have updated the follow-up to 2023.08.31.

The updated median follow-up was 24.2 months (range: 1.0–37.8 months); median OS was 27.2 months (95% CI: 20.2–NE) and median PFS was 10.6 months (95% CI: 6.4–17.8 months).

10. In the Abstract or in the Main Text, the contents that first appear should not be expressed as Abbreviation.

Reply: Full names for abbreviations have now been included in the revised manuscript.

Reviewer #2 (Remarks to the Author): with expertise in lung cancer, therapy

Zhang and colleagues provide results of a single-arm phase 2 multi-center trial conducted in China. 47 Patients received two cycles of carboplatinum, nab-paclitaxel and sintilimab for stage III/IV squamous NSCLC, followed by ICI maintenance. The design of the trial is clearly explained, methodology is sound, results are clearly reported and the conclusions drawn are comprehensible. The trial met its primary endpoint (PFS) and demonstrated encouraging efficacy (ORR, DoR) and survival results (PFS, OS). Furthermore, responses were correlated with biomarkers (blood- and tissue-based) and ct-DNA clearance.

The authors should be congratulated for having conducted this trial. ORR, PFS and OS compare very favorably to what has been reported from comparable trials (KN407, 9LA). There are some points that make the comparability to established regimens difficult. First, a relatively high share of patients with stage III disease was included (25%; 9LA/KN407: stage IV only, Empower-Lung 3 14% (sq only not reported)), and overall, the number of patients included is small.

Reply: In present trial, we enrolled 11 patients at stage III. Before their inclusion, these patients underwent a comprehensive review by a multidisciplinary tumor board. Except for 1 patient (stage IIIB) who declined radiotherapy, the remaining 10 patients were found to be unsuitable candidates for curative surgery or chest radiotherapy. The specific reasons for their ineligibility encompassed various conditions have been added to the revised manuscript as Supplementary Table 1.

Subsequently, we also conducted a comparison with a previously conducted phase 3 clinical trial involving chemotherapy combined with immunotherapy for NSCLC,

which included a certain proportion of stage III patients similar to the present trial (ORIENT-12: 21.8-24.7%; Camel-Sq: 28%; and RATIONALE 307: 31.7-36.4%). Therefore, our inclusion of 11 patients aligns with the proportions observed in previous studies.

We have added this information with appropriate citations to the revised Results section.

Supplementary Table 1 | Reasons why patients did not undergo radical surgery, neoadjuvant treatment followed by surgery or PACIFIC regimens

Number	Clinical stage	Reason
01002	IIIC	Extensive cancer-associated lymphangitis
01009	IIIB	Huge tumor mass, and the boundary between the mass and the oesophagus is unclear
01011	IIIC	High tumor burden, extensive lymph node involvement, low lung function, high risk for radiotherapy
01013	IIIC	Multiple nodules in different lung lobes on the same side
01022	IIIC	Bilateral enlargement of hilar lymph nodes, unsuitable for radiotherapy
01025	IIIC	The patient declined radiotherapy
01029	IIIB	Bilateral enlargement of hilar lymph nodes, unsuitable for radiotherapy
01030	IIIB	Satellite lesions scattering in the same lung lobe, unsuitable for radiotherapy
01034	IIIC	An extensive tumor (10.5 cm) involving left main bronchus and causing complete lung collapse
03003	IIIB	A large cystic lesion containing an air-fluid level

06001	IIIB	Recurrent left lung cancer invading the left and right main bronchus, high risk for radiotherapy
-------	------	--

Second, the estimation of the real effect size is difficult due to the single-arm design.

Reply: Yes, we agree, as a single-arm clinical trial, indeed, this is a limitation (we have already added this limitation to the revised manuscript text). We intend to conduct a randomized controlled phase 3 clinical trial in the near future to validate our findings. Initially we designated solely chemotherapy as the control group when devising the trial. However, it's worth noting that in 2019, the landscape of cancer treatment in China evolved, with chemotherapy now being complemented by immunotherapy as the new standard regime. As a result, standalone chemotherapy is no longer considered the optimal choice of treatment for patients. We therefore adjusted our trial protocol, as documented in the attached file labeled Protocol No CIBI308Y014. Due to considerations related to securing trial funding and the practical limitations on the number of eligible participants, we ultimately proceeded with a single-arm trial design. We have mentioned these points in the revised Trial Design section. It is important to realize that our intention is to conduct a randomized controlled phase 3 clinical trial in the near future. When that time comes, we will gladly welcome and incorporate your valuable suggestions, ensuring the inclusion of an appropriate control group.

Third, RECIST evaluation was done by the local investigators and was not confirmed centrally.

Reply: Although we did not centrally confirm every image of patients from local centers, remote image assessment and verification of images were conducted at baseline and critical disease progression time points.

And forth, follow-up time is quite short (15 months; only 21 PFS and 18 OS events).

Reply: Based on your recommendation, we have updated the follow-up to 2023.08.31. The updated median follow-up was 24.2 months (range: 1.0–37.8 months); median OS was 27.2 months (95% CI: 20.2–NE) and median PFS was 10.6 months (95% CI: 6.4–17.8 months). Twenty patients (45.5%) died, with a median overall survival (OS) of 27.2 months (95% CI: 20.2–NE). The 1-year OS rate was 79.1% (95% CI: 63.6–88.5%), and the 2-year OS rate was 53.3% (95% CI: 35.5–68.2%).

These data have also been updated in the revised manuscript.

Concerning baseline characteristics there is some overweight of male patients, and the share of patients with undocumented PD-L1 expression level is relatively high (>50%).

Reply: Yes, we agree. All the patients we enrolled were squamous cell lung carcinoma cases, which is the predominant lung cancer type in males and especially low in Chinese females.

Furthermore, due to the small numbers of biopsy tissue samples obtained for advanced lung squamous cell carcinoma, we could not perform PD-L1 testing on all patients. In response to the issue you raised and by Reviewer 1, we have re-analyzed the expression of PD-L1 in all tissue samples. Currently, there are PD-L1 results for

33 patients (71.7%). The remaining patients could not be tested due to the unavailability of sufficient samples.

We have addressed these concerns in the revised manuscript and added new references and data.

The nab-paclitaxel dose chosen (260 mg/m² q3w) does not represent an established dose regimen for NSCLC (at least in Europe or the U.S.) and should be commented.

Reply: We have added the following text to the revised discussion: Nab-paclitaxel can be administered weekly and every-3-weeks. A clinical study that compared the efficacy and safety of the weekly regimen to the 3-week regimen in NSCLC reported that in the intent-to-treat (ITT) population, the weekly regimen demonstrated certain advantages in terms of ORR and safety. However, there were no significant differences in PFS and OS between the two administration regimens. Subgroup analyses revealed that squamous cell carcinoma patients had better PFS and OS when treated with the 3-week regimen, showing significant statistical differences. Among these, the 260 mg/m² q3w dosage exhibited better tolerability (Socinski et al., 2010). Additionally, in a study of neoadjuvant chemotherapy combined with immunotherapy for NSCLC conducted in China, nab-paclitaxel at a dosage of 260 mg/m² q3w was also chosen for sq-NSCLC, and the patients tolerated it well (Qiu et al., 2022). Considering this information, the 260 mg/m² q3w dosage was selected.

- Socinski, M. A. et al. A dose finding study of weekly and every-3-week nab-Paclitaxel followed by carboplatin as first-line therapy in patients with advanced non-small cell lung cancer. *J. Thorac. Oncol.* 5, 852-861 (2010).

- Qiu, F. et al. Two cycles versus three cycles of neoadjuvant sintilimab plus platinum-doublet chemotherapy in patients with resectable non-small-cell lung cancer (neoSCORE): A randomized, single center, two-arm phase II trial. *J. Clin. Oncol.* 40, 8500-8500 (2022).

Whereas TRAE appear to be low, irAE are currently not reported, which I would recommend.

Reply: Thank you for the advice.

We have compiled and analyzed the irAEs recorded in the EDC system. The summary of irAEs has been added as Table 4 to the revised manuscript.

Immune-related adverse events (irAEs) of any grade were reported for 14 (30.4%) patients. The most common irAE was elevated alanine aminotransferase [6 (13.0%) patients], followed by hypothyroidism [5 (10.8%) patients], hyperthyroidism [4 (8.7%) patients] and rash [2 (4.3%) patients]. Grade 3 irAEs were observed in 2 patients, 1 with pneumonitis and the other with a rash. There were no grade 4 or 5 irAEs.

Results for biomarkers confirm the current estimation but do not add substantially new information (ct-DNA clearance is associated with favorable outcome; correlation of gene analysis with outcome is prognostic, but as the trial lacks a comparator arm no conclusions can be drawn with regard to a predictive impact). To conclude, the present investigation adds evidence for palliative first-line treatment of sq NSCLC and suggests that the number of initial cycles of chemotherapy might be limited to two.

But we cannot categorize the results with double checkpoint inhibition on the one side (CM9LA) and “classical” chemoimmunotherapy on the other side (KN407).

Reply: We agree with your suggestion and revised all statements regarding a predictive impact in the amended manuscript.

Reviewer #3 (Remarks to the Author): with expertise in biostatistics, clinical trial study design

This was an open-label single-arm phase 2 clinical trial of sintilimab, a PD-1 inhibitor, with two cycles of chemo for treatment-naive advance squamous NSCLC. The primary endpoint is PFS, with second endpoints including ORR, TTR, DOR, OS, and safety. The manuscript is well written with study design, methods and results.

1. The gender is unbalanced (only 1 female included). It’s curious why this happened.

Reply: We have added the following text to the revised Introduction and Discussion sections:

Previous studies found that squamous NSCLC (sq-NSCLC) is the major histopathological cancer type in males, with a 5.7 times higher incidence in males than in females in Chinese non-smokers (Wu et al., 2022).

The study included mostly males, which reflects the general sex disparity of sq-NSCLC in non-smoking Chinese (Wu et al., 2022). In addition, previous research indicated a stronger correlation of smoking with squamous cell cancer than with

adenocarcinoma (Kenfield et al., 2008), while smoking is extremely unpopular for Chinese women (Benedict, 2021).

- Wu, Z. et al. Sex disparity of lung cancer risk in non-smokers: a multicenter population-based prospective study based on China National Lung Cancer Screening Program. *Chin. Med. J.* 135, 1331-1339 (2022).
- Kenfield, S. A., Wei, E. K., Stampfer, M. J., Rosner, B. A. & Colditz, G. A. Comparison of aspects of smoking among the four histological types of lung cancer. *Tob. Control* 17, 198-204 (2008).
- Benedict, C. *Tobacco, Cigarettes, and Women's Status in Modern China* (Oxford Univ. Press, 2021).

2. The Sintilimab is a PD-1 inhibitor, but only half patients tested for PD-L1 and analysis was stratified by TPS<1% VS >1%, why the other half is ND (not detectable)? Why PD-L1 test is not used in screening for everyone?

Reply: Due to the small size of biopsy tissue samples in advanced lung squamous cell carcinoma, we could not perform PD-L1 testing on all patients. In response to the issue you raised and Reviewer 1, we have reanalyzed the expression of PD-L1 in all available tissue samples. Currently, there are PD-L1 results for 33 patients (71.7%). There were 19 with PD-L1 < 1%, 11 with PD-L1 expression of 1%-49% and 3 with PD-L1 ≥ 50%. The corresponding efficacy and survival analysis data have been added to the revised manuscript.

3. Line 114 said Cox univariate and multivariate analyses, but Supp Table 1 and 2 only shows multivariate analysis results, please correct.

Reply: We have corrected the term multivariate to univariate and multivariate; see new Supplemental Table 2 and Table 3.

4. Figure 3c and 3d did not specify which test was used to compare the response between ctDNA status, is the p-value from Fisher's exact test? Please describe in methods.

Reply: Yes, Fisher's exact test was used, and the corresponding statistical methods are described in detail in the manuscript.

We have added the following sentences to the body text: All statistical analyses were performed using R software (R version 4.1.2) and SAS 9.4 (SAS Institute Inc., Cary, NC). The significance with categorical variables was evaluated using Fisher's exact test. The significance with continuous variable variables was evaluated using the Wilcoxon rank sum test. The log-rank test was employed to compare PFS, OS and DOR among different PD-L1 expression levels (< 1%, 1-49%, ≥ 50%, ND). The Sankey figure was plotted using an online visualization tool (<https://sankeymatic.com/build/>). The genomic alteration figure was plotted using ComplexHeatmap package (Gu & Schlesner, 2016). The ROC curve was plotted using pROC package (Robin et al., 2011). A *P*-value < 0.05 was considered to be a statistically significant finding.

- Gu, Z., Eils, R. & Schlesner, M. Complex heatmaps reveal patterns and correlations in multidimensional genomic data. *Bioinformatics* 32, 2847-2849 (2016).

- 50. Robin, X. et al. pROC: an open-source package for R and S+ to analyze and compare ROC curves. *BMC Bioinformatics* 12, 77 (2011).

REVIEWERS' COMMENTS

Reviewer #1 (Remarks to the Author):

1. In major issue 1, the standard treatment for advanced stage NSCLC without sensitive gene mutations in my opinion is chemo-immunotherapy, including chemotherapy 4-6 cycles. The study of reducing the dosage or cycle of chemotherapy should be compared with the standard chemo-immunotherapy strategy.
2. In major issue 2, please discuss the issue that patients with high PD-L1 expression had relatively poor treatment efficacy. This is an interesting question.
3. In major issue 4, the sequencing data has not been validated through experimental means, which cannot confirm the clinical significance of this novel and valuable predictive model, which is a major flaw of this article.
4. As Supplementary Table 2 and 3 demonstrated that, I suggested that Cox regression requires that the number of independent variables with outcome events at least 15 times. So that, the model is not well applicable in this situation.

Reviewer #2 (Remarks to the Author):

The authors have made substantial amendments and all questions have been addressed adequately.

Reviewer #3 (Remarks to the Author):

Thank you for addressing the comments and incorporating the suggestions in the revision. I'm pleased to say that I'm satisfied with the changes made, and the manuscript looks stronger now.

REVIEWERS' COMMENTS

Reviewer #1 (Remarks to the Author):

1. In major issue 1, the standard treatment for advanced stage NSCLC without sensitive gene mutations in my opinion is chemo-immunotherapy, including chemotherapy 4-6 cycles. The study of reducing the dosage or cycle of chemotherapy should be compared with the standard chemo-immunotherapy strategy.

Reply: Certainly, you are correct. We have added the following detailed information to the discussion:

The present chemo-immunotherapy study with 2 cycles chemotherapy yielded results showing a median PFS time of 10.6 months (95% CI: 6.4–17.8) and the median OS time was 27.2 months [95% CI: 20.2–not evaluable (NE)], while the ORR reached 70.5%, and the DCR) was 93.2%, with a median DOR of 13.6 months (95% CI: 4.5–NE). In the KEYNOTE-407 study (chemo-immunotherapy, including 4-6 cycles chemotherapy), the median PFS was 8 months (95% CI: 6.3-8.4), median OS was 17.1 months (95% CI: 14.4-19.9), ORR was 62.6% (95% CI: 56.6-68.3), and DOR was 8.8 months (1.3+ to 28.4+) [Ref 24]. In addition, in the RATIONALE-307 study (chemo-immunotherapy, including 4-6 cycles chemotherapy), the median PFS under tislelizumab plus chemotherapy [tislelizumab + paclitaxel and carboplatin (PC), tislelizumab + nab-PC] was 7.6 months (95% CI: 6.0-9.8) and 7.6 months (95% CI: 5.8-11.0), respectively. Notably, in the same study, for patients with stage IIIB disease, the median PFS extended to 9.8 and 11.0 months, ORR was 73% (95% CI: 63.6-80.3)

for tislelizumab + PC and 75% (95% CI: 66.0-82.3) for tislelizumab + nab-PC, and DCR was 88% (95% CI: 80.2-92.8) for tislelizumab + PC and 91% (95% CI: 84.1-95.3) for tislelizumab + nab-PC [Ref 23]. Taken together, the anti-tumor effect of the present two-cycle treatment was comparable to the standard 4-6 cycles of chemotherapy combined with immunotherapy observed in other treatment regimens.

2. In major issue 2, please discuss the issue that patients with high PD-L1 expression had relatively poor treatment efficacy. This is an interesting question.

Reply: Thank you for your suggestion. We added the following text to the discussion section:

In the context of advanced NSCLC treated with mono-immunotherapy, superior efficacy in patients with high PD-L1 expression might be observed compared to those with low or negative expression. However, in the context of combination therapy, such as immunotherapy with chemotherapy, the predictive value of PD-L1 expression appears to diminish. For example, in the KEYNOTE-407 trial, it was noted that patients with PD-L1 expression ($\geq 1\%$) might experience greater benefits in terms of PFS and OS. Nevertheless, the analysis of OS found no significant difference between patients with high PD-L1 expression ($\geq 50\%$) and those with low PD-L1 expression (1–49%) (HR 0.79 vs 0.59) [Ref 24], while in the RATIONALE-307 trial, no significant correlation was found between PD-L1 expression status and PFS. Additionally, patients with high PD-L1 expression ($\geq 50\%$) exhibited higher HR values for PFS compared to those with low expression (1-49%) (tislelizumab plus PC: HR 0.50 vs 0.44; tislelizumab plus nab-PC: HR 0.43 vs 0.31) [Ref 23]. Furthermore, in various

randomized controlled trials for advanced squamous cell carcinoma of the lung, there was notable variability in the proportion of patients with high PD-L1 expression (ranging from 12.9% to 35%) [Ref 9,23], which may be linked to the higher tumor mutation burden in smoking-related squamous cell carcinoma patients.

3. In major issue 4, the sequencing data has not been validated through experimental means, which cannot confirm the clinical significance of this novel and valuable predictive model, which is a major flaw of this article.

Reply: Yes, thank you for your keen interest in our study and your valuable suggestions. The limitation you highlighted regarding our relatively small sample size is indeed a challenge for further validation through experimental means, as you rightly pointed out. We have acknowledged this constraint as a regrettable aspect in the limitation of our study in the discussion section. Actually, the biomarkers we selected in this study were designed based on some previously published research results. For example:

Tumors with *BRCA2* mutations in patients exhibit higher TMB and better OS (1-3). There is a higher TMB in *BRCA1/2* mutant NSCLC patients, but no significant OS difference was found between *BRCA1* and *BRCA2* mutants under ICI treatment (4). LSCC patients with mutant *FBXW7* have poorer survival outcomes (5). Additionally, it has been reported that the loss of *FBXW7* may increase resistance to cisplatin treatment (6). *BRINP3*, highly expressed in human osteosarcoma tissues, is negatively correlated with prognosis (7). *KIT* and *RBI* play roles as a proto-oncogene and tumor

suppressor, respectively. Mutant *RBI* in small cell lung cancer is associated with a poor prognosis (8, 9).

1. Jiang M, Jia K, Wang L, et al. Alterations of DNA damage repair in cancer: from mechanisms to applications. *Ann Transl Med.* 2020;8(24):1685. doi:10.21037/atm-20-2920
2. Samstein RM, Krishna C, Ma X, et al. Mutations in BRCA1 and BRCA2 differentially affect the tumor microenvironment and response to checkpoint blockade immunotherapy. *Nat Cancer.* 2021;1(12):1188-1203. doi:10.1038/s43018-020-00139-8
3. Zhou Z, Li M. Evaluation of BRCA1 and BRCA2 as Indicators of Response to Immune Checkpoint Inhibitors. *JAMA Netw Open.* 2021;4(5):e217728. doi:10.1001/jamanetworkopen.2021.7728
4. Chen S, Li S, Chen D, et al. Retrospective analysis of efficacy of immune checkpoint inhibitors in BRCA-mutant non-small cell lung cancer. *J Clin Oncol.* 2021;39(15_suppl):e21067-e21067. doi:10.1200/JCO.2021.39.15_suppl.e21067
5. Fan J, Bellon M, Ju M, et al. Clinical significance of FBXW7 loss of function in human cancers. *Mol Cancer.* 2022;21(1):87. doi:10.1186/s12943-022-01548-2
6. Ruiz EJ, Diefenbacher ME, Nelson JK, et al. LUBAC determines chemotherapy resistance in squamous cell lung cancer. *J Exp Med.* 2019;216(2):450-465. doi:10.1084/jem.20180742
7. Zeng W, Xu H, Wei T, Liang H, Ma X, Wang F. Overexpression of BRINP3 Predicts Poor Prognosis and Promotes Cancer Cell Proliferation and Migration via MAP4 in Osteosarcoma. *Dis Markers.* 2022;2022:2698869. doi:10.1155/2022/2698869
8. Febres-Aldana CA, Chang JC, Ptashkin R, et al. Rb Tumor Suppressor in Small Cell

Lung Cancer: Combined Genomic and IHC Analysis with a Description of a Distinct Rb-Proficient Subset. Clin Cancer Res. 2022;28(21):4702-4713. doi:10.1158/1078-0432.

9. Liu N, Wu T, Ma Y, Cheng H, Li W, Chen M. Identification and validation of RB1 as an immune-related prognostic signature based on tumor mutation burdens in bladder cancer. Anticancer Drugs. 2023;34(2):269-280. doi:10.1097/CAD.0000000000001399

4. As Supplementary Table 2 and 3 demonstrated that, I suggested that Cox regression requires that the number of independent variables with outcome events at least 15 times. So that, the model is not well applicable in this situation.

Reply: We sincerely appreciate your valuable input. Following thorough confirmation with the statistician, based on the events per variable (EPV) method, the current sample size and number of events did not meet the requirements of robustness for a multivariable analysis. Considering your suggestion, we have concurred with your recommendation to remove the multivariate analysis, opting to retain the results from the univariate analysis.